# High energy acceptor states strongly enhance exciton transfer between metal organic phosphorescent dyes

Xander de Vries[1], Reinder Coehoorn [1,2] & Peter A. Bobbert [1,3] ✉

Exciton management in organic light-emitting diodes (OLEDs) is vital for improving efficiency, reducing device aging, and creating new device architectures. In particular in white OLEDs, exothermic Förster-type exciton transfer, e.g. from blue to red emitters, plays a crucial role. It is known that a small exothermicity partially overcomes the spectral Stokes shift, enhancing the fraction of resonant donor-acceptor pair states and thus the Förster transfer rate. We demonstrate here a second enhancement mechanism, setting in when the exothermicity exceeds the Stokes shift: transfer to multiple higher-lying electronically excited states of the acceptor molecules. Using a recently developed computational method we evaluate the Förster transfer rate for 84 different donor–acceptor pairs of phosphorescent emitters. As a result of the enhancement the Förster radius tends to increase with increasing exothermicity, from around 1 nm to almost 4 nm. The enhancement becomes particularly strong when the excited states have a large spin-singlet character.

[1] Department of Applied Physics, Eindhoven University of Technology, P.O. Box 513, NL-5600 MB Eindhoven, The Netherlands. [2] Institute for Complex Molecular Systems, Eindhoven University of Technology, P.O. Box 513, NL-5600 MB Eindhoven, The Netherlands. [3] Center for Computational Energy Research, P.O. Box 6336, NL-5600 HH Eindhoven, The Netherlands. ✉email: P.A.Bobbert@tue.nl

n modern-day OLEDs, the transfer of excitons plays a crucial role. Fast exciton transfer can enhance exciton quenching and can lead to an increased efficiency roll-off at high luminance levels. In phosphorescent OLEDs, triplet diffusion affects the rate of triplet-polaron quenching (TPQ) and triplet–triplet annihilation (TTA)[1–13]. However, the transfer of excitons can also be used to enhance the device efficiency and to simplify the device architecture. This is often referred to as exciton management. Studies on phosphorescent OLEDs show that the generation of an exciton on a green emitter followed by the transfer to a red emitter can enhance the device efficiency and lifetime[14,15]. An explanation for this observation is that the fast transfer to the red emitter ensures a low exciton population on the green emitter sites, which decreases the likelihood of the exciton quenching processes mentioned above.

Exciton transfer is also used in OLEDs containing phosphorescent sensitizer molecules, from which excitons are transferred to fluorescent emitter molecules[16], or in OLEDs in which molecules showing thermally activated delayed fluorescence (TADF) act as sensitizers of fluorescent emitters (hyperfluorescence, ref. [17]). Furthermore, exciton management is used to simplify white OLED device architectures. To obtain white light, a multi-layer device is used in which excitons are mostly generated in the blue emissive layer. Subsequently, the excitons are transferred to adjacent green and red emitter layers to ensure emission from all three colors[18–24]. The process of tuning the layer thicknesses and emitter concentrations in such a device is crucial to ensure the right color balance to generate white light[24–28]. The use of computational OLED design can dramatically reduce the costs of this process. It has been shown that such computational design is actually feasible given the right mechanistic input parameters[29].

Theoretical studies on exciton transfer often focus on intramolecular Förster transfer in biological systems and polymers[30–32]. In such cases Förster transfer is in competition with "through-bond" Dexter transfer processes, since there is strong exciton–exciton wave function overlap. In contrast, in OLEDs based on small-molecule materials, intermolecular transfer is the most relevant transfer process. In a previous study, we have shown that in host-guest systems often used in OLEDs, with guest concentrations of about 10 mol% or less, Förster transfer is the dominant process[33]. In that study, we developed an efficient computational method to calculate exciton transfer, taking into account the coupling of the exciton to an arbitrary number of intramolecular vibrations in a fully quantum mechanical way. Important differences in the transfer rate occur when applying this method as compared with the commonly applied semi-classical Marcus transfer rate[33].

Here, we apply the computational method of ref. [33] to calculate the Förster transfer rate between different phosphorescent emitters, showing that this allows rapid screening of emitter molecules for favorable exciton transfer properties. The dipole–dipole type interaction leads to an $R^{-6}$ distance ($R$) dependence of the donor–acceptor (D–A) Förster transfer rate, which may be expressed as

$$k_{DA} = \frac{1}{\tau_D}\left(\frac{R_F}{R}\right)^6, \tag{1}$$

with $\tau_D$ the emissive lifetime of the donor molecule. The Förster radius $R_F$ contains the effects of the donor and acceptor orientations, the acceptor transition dipole moment and the donor–acceptor spectral overlap[34]. The spectral overlap depends on the broadening of all excited states due to the coupling to molecular vibrations and includes the overlap with higher excited states of the acceptor molecule. In ab initio calculations of the transfer rate both effects must be included. Previous studies have

shown how to include the coupling of the individual intramolecular vibrations (vibrons) to the exciton transfer[35,36]. However, most of these studies have focused on applications to crystalline molecular semiconductors and polymer systems[36,37]. To our knowledge, the inclusion of the vibronic couplings in the exothermic transfer for molecules relevant to OLED applications has not been demonstrated. In exothermic transfer, occurring between different molecules, there is a positive energy difference between the first electronically excited state of the donor and acceptor molecule. Exciton transfer to energetically higher excited states is usually not important in the case of isoenergetic transfer between like molecules[38,39], but needs to be included in the case of exothermic transfer when the donor emission energy is significantly higher than the first excited state of the acceptor. Higher excited states can then be the primary acceptor states. After transfer the exciton vibrationally relaxes from a higher state to the lowest-energy state of the acceptor.

## Results

**Exothermicity-enhanced Förster transfer.** In general, we expect the exothermic exciton transfer rate to be enhanced with respect to that for isoenergetic transfer. The reason for this is twofold, as schematically depicted in Fig. 1. Firstly, the donor and acceptor Stokes shift, which in the case of isoenergetic transfer reduces the rate, is in the case of exothermic transfer at least partially overcome by the additional energy available (Fig. 1a). Secondly, an enhancement is expected when the first excited state of the donor is in resonance with energetically higher excited states of the acceptor (Fig. 1b). For the phosphorescent iridium-cored emitters studied in this paper many of these higher excited states are expected to carry more singlet character than the three almost degenerate low-energy triplet type states. They have therefore larger transition dipole moments, leading to a large contribution to the Förster rate.

In our computational method for calculating the Förster transfer rate, we include the vibrational coupling in a manner as demonstrated in ref. [33] for the case of isoenergetic transfer. In order to properly include the higher excited states, we include now also the spin–orbit interaction. We apply the method to 84 combinations of 14 iridium-cored phosphorescent emitters commonly used in OLEDs. The emitters have been listed in Table 1 according to descending triplet energy, from 2.97 eV for $fac$-Ir(pmp)$_3$ down to 1.72 eV for NIr (triplet energies taken from literature). Our study thus spans the entire optical range, and includes emitters close to the near-ultraviolet and in the near-infrared. The molecular structures are shown in Fig. 2a and their systematic names are given in Supplementary Note 1.

The Förster transfer rate between a pair of donor (D) and acceptor (A) molecules is given by (see "Methods" section)

$$k_{DA} = \frac{2\pi}{\hbar}\frac{1}{(4\pi\epsilon_0\epsilon_r)^2}\frac{\langle\kappa^2\rangle}{R^6}\sum_{i,j}p_{D_i}\mu_{D_i}^2\mu_{A_j}^2\rho_{FC,ij}(\Delta E_{ij}),$$
$$= \frac{2\pi}{\hbar}\frac{1}{(4\pi\epsilon_0\epsilon_r)^2}\frac{\langle\kappa^2\rangle}{R^6}\int_{-\infty}^{\infty}\mu_D^2(E)\mu_A^2(E)dE, \tag{2}$$

where the sum in the first equality is over all excited donor states $D_i$, having an occupational probability $p_{D_i}$, and all excited acceptor states, $A_j$, with transition dipole moments $\mu_{D_i}$ and $\mu_{A_j}$. Vibrational effects are included in the Franck–Condon weighted density of vibrational states (FCWD) $\rho_{FC,ij}$, where $\Delta E_{ij} = E_{D_i} - E_{A_j}$ is the energy difference between the donor and acceptor excited states. For the relative dielectric constant $\epsilon_r$ we take 3, which is close to values for common OLED host materials[40–42]. Since $\epsilon_r$ only appears in the prefactor, the rates and Förster radii found in this work can be rescaled for any value of $\epsilon_r$. The factor $\langle\kappa^2\rangle$ is an orientational average of the dipole moments of the donor and

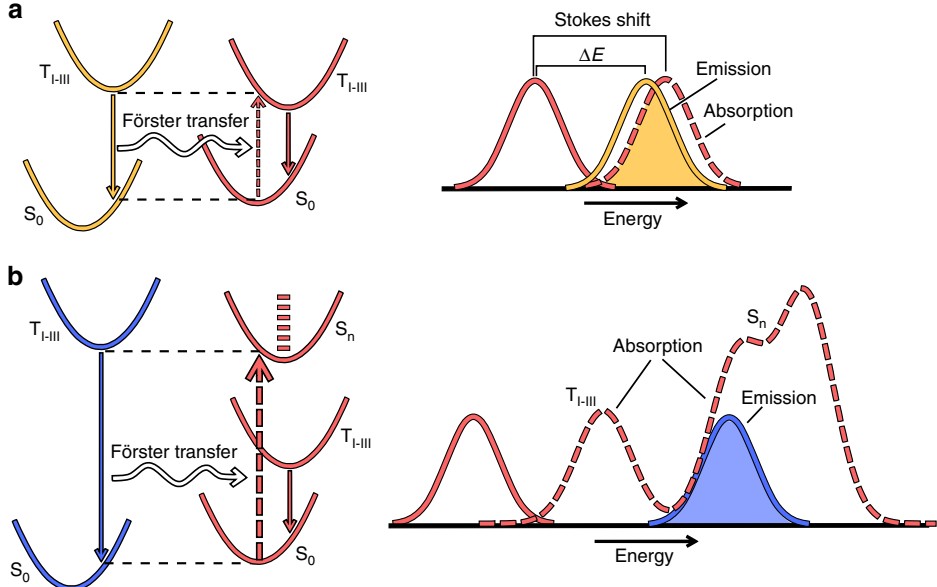

**Fig. 1 Schematic depiction of the exothermic Förster transfer mechanism studied in this work.** Transfer to emitters with lower emission energy is enhanced by the facts that: (**a**) the lowest-energy states for emission and absorption, which are predominantly of triplet character ($T_{I-III}$ states), are in resonance due to the Stokes shifts of the donor and acceptor molecules, and (**b**) the excitation can be transferred to energetically higher-lying excited acceptor states. When these states have strong singlet character ($S_n$) the transfer can be strongly enhanced.

**Table 1 Calculated and experimental triplet energies and lifetimes.**

| | | $E_T^{calc}$ | $E_T^{exp}$ | $\tau^{calc}$ | $\tau^{exp}$ |
|---|---|---|---|---|---|
| | | (eV) | (eV) | ($\mu s$) | ($\mu s$) |
| 1 | *fac*-Ir(pmp)$_3$ | 2.76 | 2.97[49] | 0.81 | 1.6[49,a] |
| 2 | *mer*-Ir(pmp)$_3$ | 2.55 | 2.70[49] | 0.55 | 1.0[49,a] |
| 3 | Firpic | 2.48 | 2.70[50] | 2.10 | 1.2[50,a] |
| 4 | Fir6 | 2.47 | 2.72[22] | 2.10 | 2.2[22] |
| 5 | Ir(ppy)$_3$ | 2.31 | 2.43[13] | 1.62 | 1.2[46,13,a] |
| 6 | Ir(ppy)$_2$acac | 2.31 | 2.38[13] | 2.10 | 1.4[13] |
| 7 | Ir(BZQ)$_2$acac | 2.16 | 2.26[51] | 1.43 | 4.5[51] |
| 8 | Ir(BT)$_2$acac | 2.12 | 2.23[13] | 2.49 | 1.3[13] |
| 9 | Ir(dpo)$_2$acac | 2.08 | 2.26[52] | 5.16 | 3.0[52] |
| 10 | Ir(npy)$_2$acac | 2.00 | 2.26[53] | 7.85 | 9.0[53] |
| 11 | Ir(MDQ)$_2$acac | 1.95 | 2.05[13] | 0.90 | 1.7[13] |
| 12 | Ir(BTP)$_2$acac | 1.88 | 2.03[46] | 9.09 | 10.8[46,a] |
| 13 | Ir(piq)$_3$ | 1.81 | 1.98[54] | 0.77 | 1.1[54] |
| 14 | NIr | 1.50 | 1.72[55] | 5.24 | – |

[a]Radiative lifetime.Otherwise: effective experimental lifetime, which is smaller than the radiative lifetime due to non-radiative decay.
*Radiative lifetime.

acceptor molecules (see "Methods" section), which is taken equal to 2/3 (random orientation).

In the second equality in Eq. (2), we have made use of the fact that the sum over donor and acceptor excited states can be expressed as an integral over the product of the squared vibrationally broadened transition dipole moment spectra of the donor and acceptor (see "Methods" section). These spectra therefore play a key role. By combining Eqs. (1) and (2) we obtain the following expression for the Förster radius:

$$R_F = \left( \frac{2\pi}{\hbar} \frac{\tau_D \langle \kappa^2 \rangle}{(4\pi\epsilon_0\epsilon_r)^2} \int_{-\infty}^{\infty} \mu_D^2(E)\mu_A^2(E)\,dE \right)^{1/6}. \quad (3)$$

This equation has structurally the same form as the equation

originally derived by Förster[34], which contains an overlap of the emission spectrum of the donor and the extinction spectrum of the acceptor. Experimentally, it is common practice to measure the emission and extinction spectra[43–46] and to determine the Förster radius from their overlap. The difficulty with that approach is that the spectra often contain tails of which the origin is not precisely known. The theoretical approach we follow in calculating the Förster radius from Eq. (3) is straightforward and unambiguous once $\mu_D^2(E)$ and $\mu_A^2(E)$ are determined.

**Triplet energies and lifetimes.** The calculated (see "Methods" section) and experimental triplet energies of the considered phosphorescent emitters are reported in Table 1 and compared in Fig. 2b. This comparison shows that the theoretical triplet energies are correctly correlated to the experimental ones, but that in most cases the experimental triplet energy is underestimated by about $0.1 - 0.2$ eV. We note that the calculations refer to the gas-phase triplet energies, whereas the experimental values are obtained from thin-film studies. Thin-film embedding, in a polarizable environment, reduces the triplet energy. The absolute error of the gas-phase calculations is thus larger than is apparent from the figure. At least part of the error can be explained from the fact that the used B3LYP DFT functional (see "Methods" section) underestimates excitation energies[47]. To our knowledge there is no detailed study of the effect on the excitation energies of different DFT functionals using the spin–orbit coupling method employed in this work. No systematic error occurs for the donor–acceptor energy difference. This is convenient, because it is this energy difference that determines the magnitude of the Förster transfer rate.

The lifetimes of the individual triplet states are calculated using the Strickler–Berg expression[48]:

$$\frac{1}{\tau_i} = \frac{4}{3} \frac{\mu_i^2 \epsilon_r}{4\pi\epsilon_0\hbar} \left( \frac{E_i}{\hbar c} \right)^3 \quad \text{with} \quad i = 1, 2, 3. \quad (4)$$

The lifetime of excitons in the low-energy triplet manifold is calculated assuming a Boltzmann distribution over the three

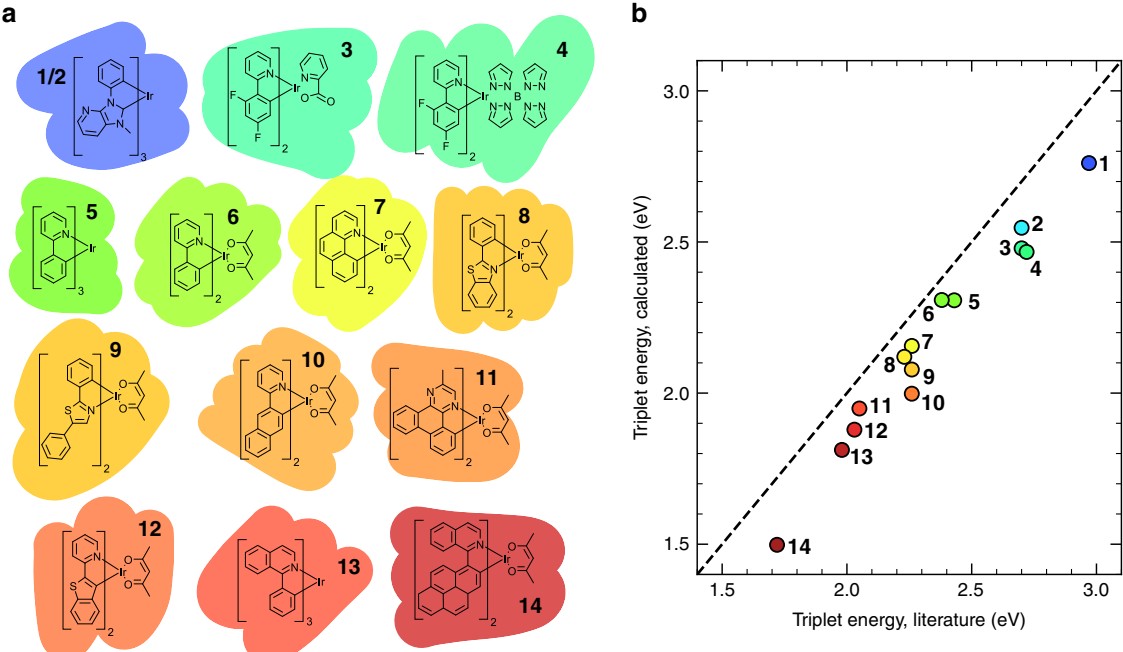

**Fig. 2 Structural formulas and triplet energies of the iridium-cored emitters studied in this work. a** Structural formulas of the iridium-cored emitters studied in this work. From left to right and top to bottom: Ir(pmp)$_3$ (*fac* (1) and *mer* (2)), FIrpic (3), FIr6 (4), Ir(ppy)$_3$ (5), Ir(ppy)$_2$acac (6), Ir(BZQ)$_2$acac (7), Ir(BT)$_2$acac (8), Ir(dpo)$_2$acac (9), Ir(npy)$_2$acac (10), Ir(MDQ)$_2$acac (11), Ir(BTP)$_2$acac (12), Ir(piq)$_3$ (13), and NIr (14). **b** Calculated triplet energies using TD-DFT (see "Methods" section) versus the experimental triplet energies reported in the literature. The symbol colors indicate the emission colors, which range from the infrared to the deep blue. The dashed line indicates $E_T^{calc} = E_T^{exp}$. The numerical values are reported in Table 1.

triplet states $T_{I-III}$[33]. All presented results from now on are for room temperature.

**Förster radii.** From Eq. (2) it follows that the Förster transfer rate is proportional to the overlap of the spectra of the squared donor and acceptor transition dipole moments $\mu_D^2(E)$ and $\mu_A^2(E)$. These spectra are calculated by summing the squared transition dipole moments of the electronically excited states, multiplied by a vibrational broadening function and, in the case of the donor spectrum, an occupational probability (see Eq. (16) in "Methods" section). As an example, Fig. 3a and b shows the spectral overlap for the green and blue emitters Ir(ppy)$_3$ and *fac*-Ir(pmp)$_3$ combined with Ir(ppy)$_3$, Ir(BT)$_2$(acac), Ir(dpo)$_2$acac, Ir(MDQ)$_2$acac, Ir(BTP)$_2$(acac), and NIr as acceptors. These six acceptor molecules follow a trend of decreasing triplet energy. The figure indeed shows that the acceptor spectra are increasingly red-shifted. In comparing the spectra in Fig. 3 with experiment one should take into account the inhomogeneous broadening due to disorder in the studied system and our slight underestimation of the triplet energies (see Fig. 2b).

Overall, we observe from Fig. 3a and b a trend of increasing Förster radius $R_F$ with increasing exothermicity $\Delta E_{DA}$, which is defined as the difference in energy between the lowest excited states of the donor and the acceptor. However, specific cases show important deviations from this trend. These deviations are related to molecule-specific structures in the spectra, which determine the spectral overlap. It is remarkable to see, for example, the change in acceptor spectra and the effect of this change on $R_F$ for transfer from Ir(ppy)$_3$ and *fac*-Ir(pmp)$_3$ to either Ir(BT)$_2$(acac) or Ir(BTP)$_2$(acac), which only differ in the position of the nitrogen atom in the BT and BTP ligands (see Fig. 2a). Our calculations show that this has a profound influence on the metal-ligand charge transfer (MLCT) character of the excitons, which is significantly larger in Ir(BT)$_2$(acac) than in Ir(BTP)$_2$(acac). As a

result, the spin–orbit coupling in Ir(BT)$_2$(acac) is much stronger than in Ir(BTP)$_2$(acac)[56]. This has an effect on the triplet lifetime, which is much shorter in Ir(BT)$_2$(acac) than in Ir(BTP)$_2$(acac) (see Table 1). In Fig. 3a (transfer from Ir(ppy)$_3$) this is seen to break the trend of increasing $R_F$ with increasing exothermicity, because the acceptor dipole moment of the triplet states in Ir(BTP)$_2$(acac) is very small (the small bump just above 2.0 eV corresponds to the $T_{III}$ state). By contrast, for the case of transfer from *fac*-Ir(pmp) in Fig. 3b, the triplet state in Ir(BTP)$_2$(acac) is irrelevant for the transfer. Transfer now occurs predominantly to a state that has large (73%) $S_1$ character (the lowest exciton singlet state without taking into account spin–orbit coupling). The situation is completely different in Ir(BT)$_2$(acac), where strong mixing by the spin–orbit coupling occurs and two excited states have large $S_1$ character (37% and 43%).

For triplet transfer from Ir(ppy)$_3$ to Ir(MDQ)$_2$(acac) in PMMA, Steinbacher et al. find a Förster radius of around 3–3.5 nm from time-resolved and steady-state photoluminescence (PL) experiments[57]. The results of our calculations are consistent with this range, when taking into account that the relative dielectric constant of PMMA is $\epsilon_r = 2$, leading to a Förster radius $R_F = 3.06$ nm instead of the value of $R_F = 2.67$ nm that we obtained for $\epsilon_r = 3$. We note that these authors also determined the Förster radius by evaluating the overlap between the measured emission spectrum of Ir(ppy)$_3$ and the absorption spectrum of Ir(MDQ)$_2$(acac) (cf. Eq. (3)), from which they find $R_F = 3.58$ nm. It would be interesting to extend such studies to more systems, to investigate more systematically the difference between the Förster radii obtained from the PL and spectral overlap methods.

Figure 4 shows a scatter plot of the Förster transfer radii $R_F$ for the 84 donor–acceptor pairs investigated in this work. The plot shows a significant spread, which is due to the molecule-specific overlap between the donor and acceptor spectra, as shown in Fig. 3 and discussed above. The Förster radii close to $\Delta E_{DA} = 0$,

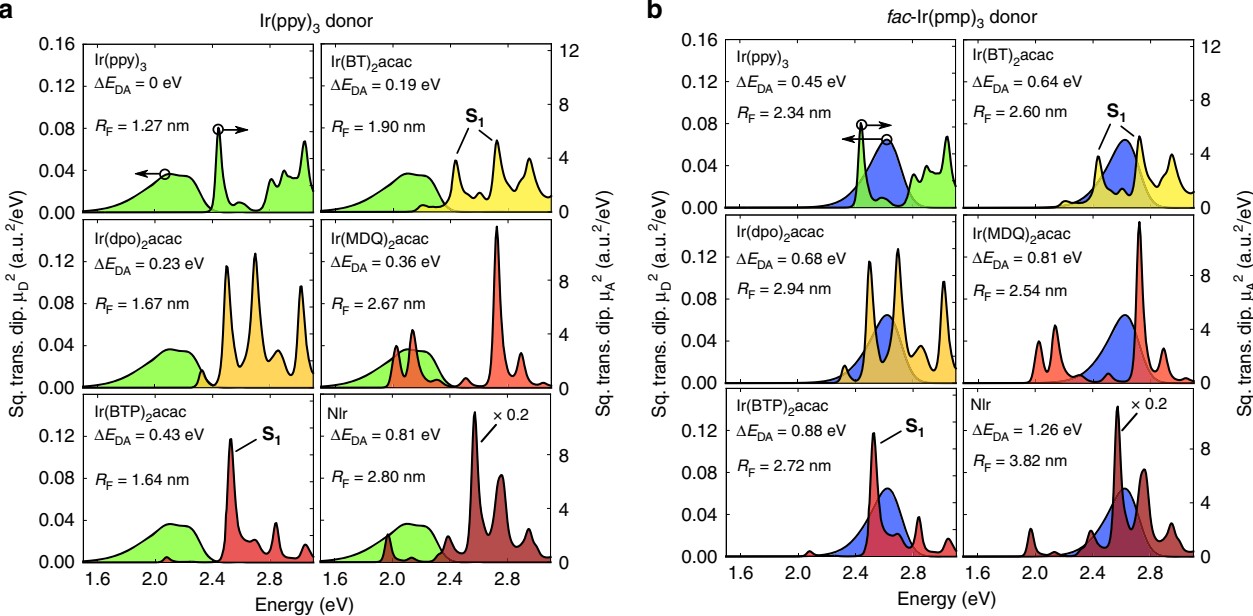

**Fig. 3 Calculated squared transition dipole moment spectra for selected donor–acceptor pairs. a, b** The Ir(ppy)$_3$ and *fac*-Ir(pmp)$_3$ donor spectrum, respectively, together with the Ir(ppy)$_3$, Ir(BT)$_2$(acac), Ir(dpo)$_2$acac, Ir(MDQ)$_2$acac, Ir(BTP)$_2$(acac), and NIr acceptor spectra. Each panel also gives the Förster radius for the exciton transfer in the corresponding donor–acceptor system. For Ir(BT)$_2$(acac) and Ir(BTP)$_2$(acac) the peaks with strong S$_1$ character are indicated.

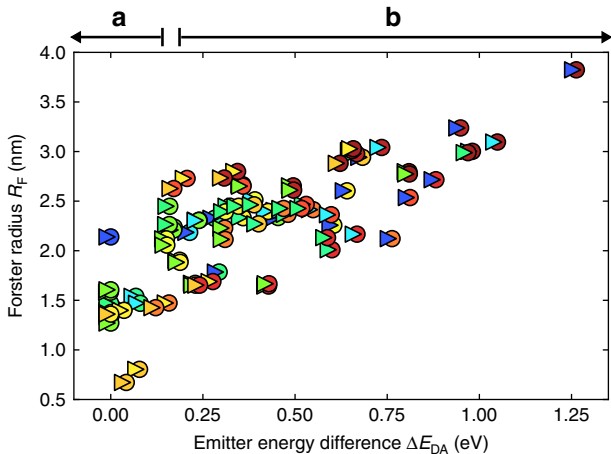

**Fig. 4 Calculated Förster radii versus energy difference for the 84 donor–acceptor combinations.** Every point in the graph represents a transfer process from a donor (triangle) to an acceptor (circle), with emission colors as indicated. The $\Delta E_{DA}$-regions indicated in the top part of the figure (a) and (b) indicate the mechanism (see Fig. 1) by which the exothermicity enhances $R_F$.

which corresponds to Förster transfer between like emitter molecules, have values of 0.7–1.5 nm, in accordance with values obtained by Kawamura et al. from concentration quenching experiments[46]. These authors also determine the Förster radii from the overlap-integral method. From the concentration quenching experiments and the overlap-integral method they find Förster radii of 1.4 and 1.8 nm in Ir(ppy)$_3$, 1.1 and 1.7 nm in FIrpic, and 0.8 and 1.5 nm in Ir(BTP)$_2$(acac), respectively. As for the case of the Ir(ppy)$_3$:Ir(MDQ)$_2$acac systems studied in ref. [57], these studies also yield a larger value of $R_F$ from the overlap-integral method.

The initial increase in Förster radius in the range $\Delta E_{DA} = 0$ $-0.15$ eV, as indicated by (a) in Fig. 4, can be explained from the

mechanism shown in Fig. 1a. For exothermic transfer larger than ~0.15 eV, the excess energy is sufficient to overcome the Stokes shift. The Förster radius then increases in general further, as indicated by (b) in Fig. 4, due to resonance of the donor states with energetically higher excited states of the acceptor, as depicted in Fig. 1b. For the largest exothermicity considered in this work (*fac*-Ir(pmp)$_3$ to NIr transfer), $R_F$ approaches a value of almost 4 nm. In Supplementary Note 5 of the SI, a similar graph is given for systems with Gaussian excitonic energy disorder, characterized by standard deviations of 0.05 and 0.10 eV. We find that such disorder has only a marginal effect on $R_F$, except when $R_F$ is very small (<1 nm) due to a poor spectral overlap. In those cases the disorder increases the spectral overlap, leading to a significant increase in $R_F$. Supplementary Note 4 of the SI contains a numerical overview of all data points, and Supplementary Fig. 2 gives separate Förster transfer graphs for every donor molecule considered. We note that endothermic exciton transfer is also possible because of thermal population of electronically and vibrationally higher-lying states of the donor with energies exceeding the energy of the first excited state of the acceptor. Such transfer will show activated behaviour and will thus, for the systems studied in this work, become unimportant when the endothermicity exceeds a few times the thermal energy.

Our analysis allows to identify the molecule-specific factors that have a large influence on the exothermic Förster transfer radius $R_F$ and those that do not. The effect of the vibronic couplings is mostly the broadening of the peaks in the spectra. Increasing vibronic couplings will therefore smoothen out the variations in $R_F$ with exothermicity, but not dramatically change the overall trend in the values of $R_F$. A change in the magnitude of the dipole moments of the donor excited states will not significantly change $R_F$, because this change will be compensated by the change in the lifetime $\tau_D$ (see Eqs. (3) and (4)). Only a change in the magnitude of the dipole moments of the acceptor excited states will significantly change $R_F$. The magnitude of the transition dipole moments of these excited states is determined by two factors: (1) firstly, by the magnitude of the pure singlet transition dipole moments of the emitter molecules,

and (2) secondly, by the strength of the spin–orbit coupling, which mixes the singlet and triplet excited states. The influence of the strength of the spin–orbit coupling has been demonstrated above in the comparison between $Ir(BT)_2(acac)$ (strong spin–orbit coupling) and $Ir(BTP)_2(acac)$ (weak spin–orbit coupling) as acceptor. It can be seen in Fig. 3 that in general higher excited states have larger transition dipole moments, which can be attributed to the increased singlet character of energetically higher-lying excitations.

## Discussion

In this study, we investigated the exothermic Förster transfer between 14 different phosphorescent emitters commonly used in OLEDs. We have used a rate expression that includes the coupling of multiple excited states of the donor and acceptor, which is necessary to correctly describe the transfer process. Quantitative predictions of energies, lifetimes and Förster radii for the different emitters and donor–acceptor transfer are given. In accordance with experimental work we find that the exothermic transfer is enhanced with respect to the isoenergetic transfer. This enhancement is explained by two effects. Firstly, the excess energy in the exothermic transfer compensates for the Stokes shift of the donor and acceptor energy levels. Secondly, at sufficient excess energy, the energetically higher excited states of the acceptor are accessible in the transfer, which further enhances the transfer rate, and particularly so when these excited states have a large singlet character. We regard these findings, together with our computational methodology to quantitatively account for the higher excited acceptor states, as valuable contributions to the important research on Förster energy transfer among phosphorescent emitters.

The rate expression Eq. (2) and the calculations performed in this work can serve as input for device models that include energetic, spatial, and orientational disorder. This paves the way for in silico design and optimization of, for example, white OLED device structures[18–24] and TADF sensitizer OLED architectures[16,17]. For the emitters studied in this work, we have provided the squared transition dipole moment spectra. The overlap of these spectra can be numerically integrated to yield Förster radii for all the emitter combinations. These results can then be directly applied in OLED device simulations.

We have shown how the Förster radius for exciton transfer is sensitively determined by the electronic and vibronic interactions that determine the nature of the excitonic states on the emitter molecules. Designing novel phosphorescent emitters for improved exciton management in OLEDs, i.e., for more favorable exciton transfer, will thus require better understanding and tuning of the structural motifs that affect these interactions.

## Methods

**Calculation of excited state energies and transition dipole moments.** The excited state energies and transition dipole moments are calculated using time-dependent density functional theory (TD-DFT), including spin–orbit coupling in the zeroth-order regular approximation (ZORA) to the Dirac equation. The calculations for the donor molecules are done in the spin-restricted DFT triplet geometry, while the calculations for the acceptor are done in the density functional theory (DFT) ground state geometry. All calculations have been performed with the B3LYP functional, a segmented all-electron relativistically controlled (SARC) basis set for the iridium atom and the 6-31G basis set for the other atoms. For the DFT calculations the scalar relativistic ZORA correction was included. All methods are available in the used ORCA package[58–60].

**Expressions for the Förster transfer rate.** Taking into account all excited states of the donor, $D_i$, with thermal occupation probability $p_{D_i}$, and all excited states of the acceptor, $A_j$, Fermi's golden rule for the exciton transfer from donor

to acceptor yields

$$k_{DA} = \sum_{i,j} p_{D_i} \frac{2\pi}{\hbar} J_{ij}^2 \rho_{FC,ij}(\Delta E_{ij}),$$ (5)

with $\Delta E_{ij} = E_{D_i} - E_{A_j}$ the energy difference between the donor and acceptor excited states. Here, $J_{ij}$ is the transfer integral,

$$J_{ij} = \frac{1}{4\pi\epsilon_0\epsilon_r} \frac{\kappa_{ij}\mu_{D_i}\mu_{A_j}}{R^3},$$ (6)

with $\epsilon_0$ the vacuum dielectric permittivity, $\epsilon_r$ the relative dielectric constant, $\kappa_{ij}$ an orientational factor, and $\mu_{D_i}$ and $\mu_{A_j}$ the transition dipole moments of the donor and acceptor states[30]. In Eq. (6), $\rho_{FC,ij}$ is the Franck–Condon weighted density of vibrational states (FCWD), which is a sum over all possible combinations of vibronic transitions of the donor molecule and the acceptor molecule in the excited states $D_i$ and $A_j$. The calculation of $\rho_{FC,ij}$ is discussed below. Combining Eq. (5) with Eq. (6) and performing an average over $\langle\kappa_{ij}^2\rangle$ yields the first equality in Eq. (2) in the main text for the Förster transfer rate.

For the phosphorescent donor molecules discussed in this work, we need to consider in practice only the first three excited states (the first triplet manifold $T_{I–III}$). The energetic separation between these states is at room temperature smaller than or close to $k_BT$, while the separation with other states is often much larger than $k_BT$. The total occupational probability of these first three donor states is therefore often very close to 1, so that the $i$-summation in Eq. (5) can be restricted to these states. For Förster transfer between identical molecules, the terms in the $j$-sum in Eq. (5) will quickly become negligibly small because transfer to the higher excited states of the acceptor molecule requires that $\Delta E_{ij} < 0$. Such endothermic transfer is unlikely because the $\rho_{FC,ij}$ factor in Eq. (5) scales with $\exp(\Delta E_{ij}/k_BT)$ for $\Delta E_{ij} < 0$ and the rate therefore decreases exponentially.

We calculate the FCWD $\rho_{FC,ij}$ using the displaced harmonic oscillator model and the parallel mode approximation[36,61,62]. In this model, we assume that the normal mode coordinates of the final excitonic state are shifted with respect to the normal mode coordinates in the initial excitonic state. The normal mode displacement vectors in mass-weighted coordinates, $\mathbf{K}_{n_i}$ and $\mathbf{K}_{n_j}$, are calculated as explained in detail in Supplementary Note 2. These vectors are used to obtain the donor and acceptor vibronic coupling parameters:

$$\lambda_{n_i} = \frac{1}{2}K_{n_i}^2\omega_{n_i}^2 \quad \text{and} \quad \lambda_{m_j} = \frac{1}{2}K_{m_j}^2\omega_{m_j}^2,$$ (7)

where $\omega_{n_i}$ is the vibron frequency of mode $n_i$ of excited state $i$ of the donor and $\omega_{m_j}$ is the vibron frequency of mode $m_j$ of excited state $j$ of the acceptor. The FCWD $\rho_{FC,ij}$ involves the coupling of the exciton to the donor vibron modes of excited state $i$ and acceptor vibron modes of excited state $j$. We use a time integral to evaluate $\rho_{FC,ij}$ at $\Delta E_{ij}$:

$$\rho_{FC,ij}(\Delta E_{ij}) = \frac{1}{2\pi\hbar}\int_{-\infty}^{\infty} \exp(i\Delta E_{ij}t)\rho_{FC,ij}(t)dt,$$ (8)

with

$$\rho_{FC,ij}(t) = I_{D_i}(t)I_{A_j}(t),$$ (9)

where we have factorized $\rho_{FC,ij}(t)$ into a donor contribution

$$I_{D_i}(t) = \exp\left(-i\sum_{n_i} \frac{K_{n_i}^2\omega_{n_i}}{\cot\left[\frac{\omega_{n_i}t}{2}\right] - \cot\left[\frac{\omega_{n_i}(t+i/k_BT)}{2}\right]}\right),$$ (10)

and an acceptor contribution

$$I_{A_j}(t) = \exp\left(-i\sum_{m_j} \frac{K_{m_j}^2\omega_{m_j}}{\cot\left[\frac{\omega_{m_j}t}{2}\right] - \cot\left[\frac{\omega_{m_j}(t+i/k_BT)}{2}\right]}\right).$$ (11)

Equations (10) and (11) are equivalent but more compact forms of an equation used in our previous work[62], based on the work by de Souza et al.[63]. Using the convolution theorem we can also write Eq. (8) as the overlap of the emission spectrum of excited state $i$ of the donor and the absorption spectrum of excited state $j$ of the acceptor:

$$\rho_{FC,ij}(\Delta E_{ij}) = \int_{-\infty}^{\infty} I_{D_i}(E)I_{A_j}(E)dE,$$ (12)

where

$$I_{D_i}(E) = \frac{1}{2\pi\hbar}\int_{-\infty}^{\infty} \exp[i(E_{D_i}-E)t]I_{D_i}(t)dt,$$ (13)

and

$$I_{A_j}(E) = \frac{1}{2\pi\hbar}\int_{-\infty}^{\infty} \exp[i(E-E_{A_j})t]I_{A_j}(t)dt.$$ (14)

The excited state energies of the molecules are calculated with TD-DFT for the ground state molecular geometry. These energies are the "vertical" excitation

energies, $E_{\mathrm{D}_i, \mathrm{A}_j}^{\mathrm{vert}}$. The adiabatic excitation energies include the reorganization energies related to the geometrical relaxation that happens in the excited states:

$$E_{\mathrm{D}_i}^{\mathrm{adia}} = E_{\mathrm{D}_i}^{\mathrm{vert}} + \sum_{n_i} \lambda_{n_i} \quad \text{and} \quad E_{\mathrm{A}_j}^{\mathrm{adia}} = E_{\mathrm{A}_j}^{\mathrm{vert}} - \sum_{m_j} \lambda_{m_j}. \tag{15}$$

In the calculation of the Förster transfer rate we only consider the first three triplet states of the donor (see above). The triplet energy of the donor is taken to be the that of the lowest triplet state $T_I$ of the triplet manifold ($E_T = E_{\mathrm{D}_I}^{\mathrm{adia}}$). No significant change in the final results occurs when selecting one of the two other states as the reference level, because the three triplets $T_{\mathrm{I-III}}$ are very close in energy. For the acceptor excited states we consider all excitations up to 3.1 eV. This is 0.35 eV higher than the triplet energy of *fac*-Ir(pmp)$_3$, which has the highest emission energy at 2.76 eV. In addition, we consider only excited states with transition dipole moments larger than 1% of the sum over all excited state transition dipole moments. This limits the computational effort with a negligible loss of accuracy.

We define the squared transition dipole spectra of the donor and acceptor as

$$\mu_{\mathrm{D}}^2(E) = \sum_i p_{\mathrm{D}_i} \mu_{\mathrm{D}_i}^2 I_{\mathrm{D}_i}(E) \quad \text{and} \quad \mu_{\mathrm{A}}^2(E) = \sum_j \mu_{\mathrm{A}_j}^2 I_{\mathrm{A}_j}(E). \tag{16}$$

Combining Eqs. (12) and (16) we obtain

$$\sum_{i,j} p_{\mathrm{D}_i} \mu_{\mathrm{D}_i}^2 \mu_{\mathrm{A}_j}^2 \rho_{\mathrm{FC},ij}(\Delta E_{ij}) = \int_{-\infty}^{\infty} \mu_{\mathrm{D}}^2(E) \mu_{\mathrm{A}}^2(E) dE. \tag{17}$$

This yields the second equality in Eq. (2) in the main text for the Förster transfer rate. In Supplementary Note 3 in the SI we provide the squared transition dipole spectra for all 14 emitter molecules studied in this work. These can be used to calculate the donor–acceptor transfer rate between any combination of molecules. In Eqs. (8), (13) and (14)″, we applied a Gaussian broadening of 10 meV, well below the inhomogeneous broadening expected for realistic materials, to the energy levels to allow for numerical integration.

## Data availability
Data supporting this publication is available from the corresponding author on request. The (TD-)DFT calculations were performed using the ORCA package, available at orcaforum.kofo.mpg.de. The calculations of the transfer rates were performed with in house developed software. The calculated squared donor and acceptor dipole moment spectra of the 14 studied emitters are available from the corresponding author on request.

## Code availability
The (TD-)DFT calculations were performed using the ORCA package, available at orcaforum.kofo.mpg.de. The calculations of the transfer rates were performed with in house developed software which is available from the corresponding author on request.

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

## Acknowledgements

This research is part of the Horizon-2020 EU project MOSTOPHOS (Project No. 646259).

## Author contributions

X.d.V. performed all the calculations presented in this work. R.C. and P.A.B. guided the work. All authors participated in discussions about the work and contributed to the writing of the manuscript.

## Competing interests

The authors declare no competing interests.
