## [Peer Review File · Nature Communications]

Reviewers' comments:

Reviewer #1 (Remarks to the Author):

In the present manuscript, the authors present an in-depth study of exciton transfer between phosphorescent organometallic dyes based on a computational methodology. The work presented is carried out with very high quality and addresses one of the most central processes in organic optoelectronic devices. In particular, the relative energy position of donor and acceptor states is investigated and it is shown that the energy transfer efficiency does increase substantially, if the energy difference between donor and acceptor states increases, so that higher lying acceptor states contribute to the overall process. In general, this study is an excellent addition to the current understanding and I assume that it will be welcomed by the research community. Hence, I am in favor for its publication in Nature Communication. Only, there are some points, which I think should be addressed to sharpen this work to make it accessible to a very broad audience. Please find these points subsequently:

1. The terminology used in the manuscript can be slightly misleading. In particular, there are two points, which I would like to mention:

a) One key result is that the energy transfer becomes more effective, when the energy splitting between donor and acceptor triplet states increases. Here, the authors speak about exothermic transfer to describe exactly this setting (large difference between donor and acceptor states). In contrast, they denote a transfer as isoenergetic, if donor and acceptor states are equal (similar). However, looking at the energy transfer in general, it is a two step process: (i) the transfer of an 'energy quant' of a donor to the acceptor. This step is actually conserving the energy and hence, isoenergetic. As a consequence, the energy populates a state of the acceptor within its vibronic energy spectrum. (ii) In a second step, the energy may thermalize to the lower excited electronic state (losing vibrational quanta). Consequently, overall this process may lose energy (which is why the authors categorize it as exothermic), but the actual process of energy transfer is isoenergetic for each transition within the overlap of donor and acceptor states.

b) The authors refer in their manuscript, that the transfer to higher excited states of the acceptor helps to increase the transfer efficiency or range. Here, I would suggest the authors to sharpen their terminology. 'higher excited state' can be anything (a higher electronic level, a higher vibrational level of a given electronic level, etc.) I assume that in particular the admixture of higher electronic states in the acceptor spectrum is important for the effects observed.

2. In Figure 3, the the calculated squared transition dipole moment spectra are shown for the various donor and acceptor cases. First, as a general comment, I would like to suggest to increase the description of how these spectra are obtained. Are they solely obtained from computation? In particular, the respective donor spectra should be easy to be compared to experimental spectra (PL spectra of the donor). Or is the donor spectrum based on the PL spectra of the respective compound? For the acceptor spectra, I assume that the transitions originate from computation which are then broadened to realize these final spectra. In particular for the experimental research community, it will be very helpful, if the authors can detail the way to obtain these spectra.

3. One more point to the data in Figure 3: It would be helpful, if the authors could include the assignment of μ_D and μ_A in the plots, so that it is consistent with Eqns. 2 and 3.

4. What comes to mind when looking at the many combinations of emitters the authors present is the question, whether energy transfer between nominally negative energy differences (donor level slightly below acceptor) is possible. I come across this question, because for the case of like emitters (donor and acceptor the same molecule), there is a non-zero Förster radius. E.g. Ir(ppy)₃ 1.27 nm. How much 'endothermicity' - staying with the terms of the authors - can there be and still energy transfer is possible?

5. There is a typo in the Methods section: page 14: 'and all excited TATES of the acceptor'

Reviewer #2 (Remarks to the Author):

The manuscript reports calculations on the Förster transfer rate for a series of phosphorescent emitters whose emission ranges from the very red to the very blue part of the spectrum. In a thorough, systematic study the authors find the Förster radius to increase with the energy difference between donor and acceptor, in particular when transfer can occur to higher-lying singlet states of the acceptor.

I like the paper – it reads extremely well, and it is a beautifully and thoroughly conducted study. What I was, at first, not sure about is whether in its current form it represents “an important advance of significance to specialists in the field”. The reported result, i.e. that Förster transfer can occur to higher singlets and in this way lead to a larger Förster radius, is in full agreement with our understanding on Förster-type energy transfer and thus, while very nicely demonstrated, is, at first sight, not advancing current knowledge. At second sight, it is remarkable that this process can be applied so well to phosphorescent compounds in which the nominal triplet states have a mixed character, and that the theoretical methodology seems to work. The authors give comparison between experimental and theoretical energies and lifetimes. Here, I would have appreciated if the authors could have included a comparison between the Förster radius they obtain in their calculations and the Förster radius that is obtained when evaluating the spectral overlap integral from emission and absorption, at least in a few selected cases. Overall, I recommend publication, though I suggest that the authors emphasize the novel aspects of their study somewhat stronger in their conclusions so that they are also apparent also when reading swiftly.

Reviewer #1 (Remarks to the Author)

In the present manuscript, the authors present an in-depth study of exciton transfer between phosphorescent organometallic dyes based on a computational methodology. The work presented is carried out with very high quality and addresses one of the most central processes in organic optoelectronic devices. In particular, the relative energy position of donor and acceptor states is investigated and it is shown that the energy transfer efficiency does increase substantially, if the energy difference between donor and acceptor states increases, so that higher lying acceptor states contribute to the overall process. In general, this study is an excellent addition to the current understanding and I assume that it will be welcomed by the research community. Hence, I am in favor for its publication in Nature Communication. Only, there are some points, which I think should be addressed to sharpen this work to make it accessible to a very broad audience. Please find these points subsequently:

1. The terminology used in the manuscript can be slightly misleading. In particular, there are two points, which I would like to mention:

a) One key result is that the energy transfer becomes more effective, when the energy splitting between donor and acceptor triplet states increases. Here, the authors speak about exothermic transfer to describe exactly this setting (large difference between donor and acceptor states). In contrast, they denote a transfer as isoenergetic, if donor and acceptor states are equal (similar). However, looking at the energy transfer in general, it is a two step process: (i) the transfer of an 'energy quant' of a donor to the acceptor. This step is actually conserving the energy and hence, isoenergetic. As a consequence, the energy populates a state of the acceptor within its vibronic energy spectrum. (ii) In a second step, the energy may thermalize to the lower excited electronic state (losing vibrational quanta). Consequently, overall this process may lose energy (which is why the authors categorize it as exothermic), but the actual process of energy transfer is isoenergetic for each transition within the overlap of donor and acceptor states.

The reviewer is right that this can be slightly misleading. We still would like to stick to the terminology "exothermic" and "isoenergetic", because this complies with literature (see e.g. Ref. 50). In order to prevent any misunderstanding, we changed the last sentences of the second paragraph on p. 3 to (additions underlined):

"To our knowledge, the inclusion of the vibronic couplings in the exothermic transfer for molecules relevant to OLED applications has not been demonstrated. In exothermic transfer, occurring between different molecules, there is a positive energy difference between the first electronically excited state of the donor and acceptor molecule. Exciton transfer to energetically higher excited states is usually not important in the case of isoenergetic transfer between like molecules [38,39], but needs to be included in the case of exothermic transfer when the donor emission energy is significantly higher than the first excited state of the acceptor. Higher excited states can then be the primary acceptor states. After transfer the exciton vibrationally relaxes from a higher state to the lowest-energy state of the acceptor."

b) The authors refer in their manuscript, that the transfer to higher excited states of the acceptor helps to increase the transfer efficiency or range. Here, I would suggest the authors to sharpen their terminology. 'Higher excited state' can be anything (a higher electronic level, a higher vibrational level of a given electronic level, etc.) I assume that in particular the admixture of higher electronic states in the acceptor spectrum is important for the effects observed.

We agree that this can also be slightly misleading. In order to prevent any misunderstanding, we changed in the abstract of the paper "transfer to multiple higher excited states of the acceptor molecules" to "transfer to multiple higher-lying electronically excited states of the acceptor molecules". As seen in the added sentences mentioned under a) we now also talk about the "first electronically excited state of the donor and

acceptor molecule". Later on, we assume that the reader understands that we mean "electronically excited states" when we talk about "excited states".

2. In Figure 3, the calculated squared transition dipole moment spectra are shown for the various donor and acceptor cases. First, as a general comment, I would like to suggest to increase the description of how these spectra are obtained. Are they solely obtained from computation? In particular, the respective donor spectra should be easy to be compared to experimental spectra (PL spectra of the donor). Or is the donor spectrum based on the PL spectra of the respective compound? For the acceptor spectra, I assume that the transitions originate from computation which are then broadened to realize these final spectra. In particular for the experimental research community, it will be very helpful, if the authors can detail the way to obtain these spectra.

Following the suggestion of the reviewer, we have added more detail to the description of how the spectra in Figure 3 were obtained; see the following sentence added to the first paragraph under "Förster radii" on p. 7:

"These spectra are calculated by summing the squared transition dipole moments of the electronically excited states, multiplied by a vibrational broadening function and, in the case of the donor spectrum, a Boltzmann state occupation factor (see Eq. (16) in Methods)."

These spectra are thus solely obtained from computation, which should now be clear from the added sentence. Indeed, the spectra can be compared to experiment. What then needs to be taken into account are the inhomogeneous broadening due to disorder and our slight underestimation of the triplet energies. To make this clear we also added to the same paragraph as above the sentence

"In comparing the spectra in Fig. 3 with experiment one should take into account the inhomogeneous broadening due to disorder in the studied system and our slight underestimation of the triplet energies (see Fig. 2(b))."

We note that we studied the effect of inhomogeneous broadening on the values of the Förster radii; see Fig. S3 in the SI. Finally, it might not have been clear that, like in the work of de Souza et al. [63], we apply a small Gaussian broadening to the energy levels to enable the numerical evaluation of the integrals in Eqs. (8), (13) and (14). To make this clear, we added to the end of the Methods section the sentence

"In Eqs. (8), (13) and (14) we applied a Gaussian broadening of 10 meV, well below the inhomogeneous broadening expected for realistic materials, to the energy levels to allow for numerical integration."

3. One more point to the data in Figure 3: It would be helpful, if the authors could include the assignment of μ_D and μ_A in the plots, so that it is consistent with Eqs. 2 and 3.

We followed the suggestion of the reviewer by adding subscripts "D" and "A" to the axis titles. See the revised Figure 3.

4. What comes to mind when looking at the many combinations of emitters the authors present is the question, whether energy transfer between nominally negative energy differences (donor level slightly below acceptor) is possible. I come across this question, because for the case of like emitters (donor and acceptor the same molecule), there is a non-zero Förster radius. E.g. Ir(ppy)₃ 1.27 nm. How much 'endothemicity' - staying with the terms of the authors - can there be and still energy transfer is possible?

It is indeed an interesting question for how much endothermicity energy transfer is still possible. The formalism developed in this work can also be used for such a case. The result is simply a fast decrease of the Förster rate with increasing endothermicity, leading to Förster radii well below the intermolecular distance when the endothermicity exceeds a few times the thermal energy. We added the following short discussion about endothermic energy transfer to the second paragraph on p. 11:

“We note that endothermic exciton transfer is also possible because of thermal population of electronically and vibrationally higher-lying states of the donor with energies exceeding the energy of the first excited state of the acceptor. Such transfer will show activated behaviour and will thus, for the systems studied in this work, become unimportant when the endothermicity exceeds a few times the thermal energy.”

5. There is a typo in the Methods section: page 14: 'and all excited TATES of the acceptor'

This has been corrected. We thank the reviewer for noting this.

Reviewer #2 (Remarks to the Author):

The manuscript reports calculations on the Förster transfer rate for a series of phosphorescent emitters whose emission ranges from the very red to the very blue part of the spectrum. In a thorough, systematic study the authors find the Förster radius to increase with the energy difference between donor and acceptor, in particular when transfer can occur to higher-lying singlet states of the acceptor.

I like the paper – it reads extremely well, and it is a beautifully and thoroughly conducted study. What I was, at first, not sure about is whether in its current form it represents “an important advance of significance to specialists in the field”. The reported result, i.e. that Förster transfer can occur to higher singlets and in this way lead to a larger Förster radius, is in full agreement with our understanding on Förster-type energy transfer and thus, while very nicely demonstrated, is, at first sight, not advancing current knowledge. At second sight, it is remarkable that this process can be applied so well to phosphorescent compounds in which the nominal triplet states have a mixed character, and that the theoretical methodology seems to work.

We thank the reviewer for the compliments about our paper. We agree that Förster transfer to higher-lying singlet states of the acceptor is in agreement with current knowledge, but that the fact that this idea and the theoretical methodology can also be successfully applied to phosphorescent compounds, with all the added complications of spin-orbit coupling, is remarkable.

The authors give comparison between experimental and theoretical energies and lifetimes. Here, I would have appreciated if the authors could have included a comparison between the Förster radius they obtain in their calculations and the Förster radius that is obtained when evaluating the spectral overlap integral from emission and absorption, at least in a few selected cases. Overall, I recommend publication, though I suggest that the authors emphasize the novel aspects of their study somewhat stronger in their conclusions so that they are also apparent also when reading swiftly.

Regarding a comparison between different ways to obtain the Förster radius of the energy transfer between donor and acceptor molecule we would like to remark that the theoretical Förster radius describing transfer at the molecular scale, as we calculate it from Eq. (3), comes from an overlap integral of the squared dipole moment spectra of the donor and

the acceptor, which is equivalent to an overlap integral of the emission and absorption spectra. However, experimentally, the Förster radius can be obtained in two ways: (1) by actually deducing the energy transfer rate at the molecular scale from a time-resolved photoluminescence experiment, and (2) from a measurement of the overlap between the emission and absorption spectra. The latter approach is more indirect, and could lead to a different (larger) value of the Förster radius in the case of inhomogeneously broadened spectra of materials with spatially correlated exciton energies. In the (scarcely available) literature, both methods have been used. For all the donor-acceptor combinations we studied, we could only make a comparison to the work of Steinbacher et al. [57] (exothermic transfer from Ir(ppy)₃ to Ir(MDQ)₂(acac)) and Kawamura et al. [46] (isoenergetic transfer in between Ir(ppy)₃, Flrpic, and Ir(BTP)₂(acac) emitters). In these studies, the Förster radius is determined in both ways indicated above (directly and from the overlap integral). We have in the paper compared our calculated results for the Förster radius to the direct measurements of the energy transfer in these two studies, because we think that the directly measured results are more reliable than the “overlap” results. Since it is indeed also interesting to report the “overlap” results we now do so. For this reason, we changed the last paragraph starting on p. 9 to

“For triplet transfer from Ir(ppy)₃ to Ir(MDQ)₂(acac) in PMMA, Steinbacher et al. find a Förster radius of around 3 - 3.5 nm from time-resolved and steady-state photoluminescence (PL) experiments [57]. The results of our calculations are consistent with this range, when taking into account that the relative dielectric constant of PMMA is $\epsilon_r = 2$, leading to a Förster radius $R_F = 3.06$ nm instead of the value of $R_F = 2.67$ nm that we obtained for $\epsilon_r = 3$. We note that these authors also determined the Förster radius by evaluating the overlap between the measured emission spectrum of Ir(ppy)₃ and the absorption spectrum of Ir(MDQ)₂(acac) (cf. Eq. (3)), from which they find $R_F = 3.58$ nm. It would be interesting to extend such studies to more systems, to investigate more systematically the difference between the Förster radii obtained from the PL and spectral overlap methods.

We also changed the last paragraph starting on p. 10 to

“Figure 4 shows a scatter plot of the Förster transfer radii R_F for the 84 donor-acceptor pairs investigated in this work. The plot shows a significant spread, which is due to the molecule specific overlap between the donor and acceptor spectra, as shown in Fig. 3 and discussed above. The Förster radii close to $\Delta E_{DA} = 0$, which correspond to Förster transfer between like emitter molecules, have values of 0.7 - 1.5 nm, in accordance with values obtained by Kawamura et al. from concentration quenching experiments [46]. These authors also determine the Förster radii from the overlap-integral method. From the concentration quenching experiments and the overlap-integral method they find Förster radii of 1.4 and 1.8 nm in Ir(ppy)₃, 1.1 and 1.7 nm in Flrpic, and 0.8 and 1.5 nm in Ir(BTP)₂(acac), respectively. As for the case of the Ir(ppy)₃:Ir(MDQ)₂(acac) systems studied in Ref. 57, these studies also yield a larger value of R_F from the overlap-integral method.”

We followed the suggestion of the reviewer to emphasize the novel aspects of our study somewhat stronger by adding the following sentence to the first paragraph of the “Summary and outlook” section on p. 13:

“We think that these findings, together with our computational methodology to quantitatively account for the higher excited acceptor states, are valuable novel contributions to the important research on Förster energy transfer among phosphorescent emitters.”

REVIEWERS' COMMENTS:

Reviewer #1 (Remarks to the Author):

In the present revision NCOMMS-19-19567B, the authors have carefully considered and addressed all the points I raised during the initial review of their manuscript. I do not see open ends and am confident to recommend this present revision for acceptance.

Sebastian Reineke

Reviewer #2 (Remarks to the Author):

The authors have satisfactorily addressed the concerns of both reviewers. No further revisions are needed. I recommend publication now.